# A novel perovskite oxide chemically designed to show multiferroic phase boundary with room-temperature magnetoelectricity

Carmen M. Fernández-Posada[1], Alicia Castro[1], Jean-Michel Kiat[2,3], Florence Porcher[3], Octavio Peña[4], Miguel Algueró[1] & Harvey Amorín[1]

There is a growing activity in the search of novel single-phase multiferroics that could finally provide distinctive magnetoelectric responses at room temperature, for they would enable a range of potentially disruptive technologies, making use of the ability of controlling polarization with a magnetic field or magnetism with an electric one (for example, voltage-tunable spintronic devices, uncooled magnetic sensors and the long-searched magnetoelectric memory). A very promising novel material concept could be to make use of phase-change phenomena at structural instabilities of a multiferroic state. Indeed, large phase-change magnetoelectric response has been anticipated by a first-principles investigation of the perovskite $BiFeO_3$–$BiCoO_3$ solid solution, specifically at its morphotropic phase boundary between multiferroic polymorphs of rhombohedral and tetragonal symmetries. Here, we report a novel perovskite oxide that belongs to the $BiFeO_3$–$BiMnO_3$–$PbTiO_3$ ternary system, chemically designed to present such multiferroic phase boundary with enhanced ferroelectricity and canted ferromagnetism, which shows distinctive room-temperature magnetoelectric responses.

[1] Instituto de Ciencia de Materiales de Madrid, CSIC. Cantoblanco, 28049 Madrid, Spain. [2] Laboratoire Structures, Propriétés et Modélisation des Solides, Associé au CNRS (UMR8580), Ecole Centrale Paris, 92295 Chatenay-Malabry, France. [3] Laboratoire Léon Brillouin, UMR 12 CEA-CNRS, CEA/Saclay, 91991 Gif-Sur-Yvette Cedex, France. [4] Institut des Sciences Chimiques de Rennes, Associé au CNRS (UMR 6226), Université de Rennes 1, 35042 Rennes, France. Correspondence and requests for materials should be addressed to H.A. (email: hamorin@icmm.csic.es).

Magnetoelectric multiferroics are compounds that show coexistence of electrical and magnetic orders, and a range of coupling phenomena between polarization and magnetization. They are highly topical materials from both the fundamental and applied points of view[1]. The understanding of the mechanisms behind the magnetoelectric coupling is a major line of research in solid state physics that must result in the development of novel approaches for obtaining functional responses, either linear magnetoelectric effect or crossed switching phenomena at room temperature[2]. This is required for enabling a number of potentially disruptive magnetoelectric technologies that would make use of the possibility of controlling polarization with a magnetic field or magnetism with an electric one, such as electrically tunable microwave components and spintronic devices[3], and the long-searched electrical-writing magnetic-reading random access memory[4].

Among the various families of multiferroic compounds under study, magnetic (or spin-induced) ferroelectrics have concentrated a lot of attention. They are improper ferroelectrics like orthorhombic $TbMn_2O_5$, in which a small (below $1 \mu C cm^{-2}$) spontaneous polarization develops as a byproduct of a magnetic transition, often involving complex incommensurate antiferromagnetic states that are intrinsically associated with the existence of spin frustration due to competing interactions. Though large magnetoelectric responses have been reported as a result of the inherent relationship between the electrical and magnetic orders, by tuning the ferroelectric state with a magnetic field, this is usually only possible at very-low temperatures, and the mechanism does not allow for a comparable electrical control of magnetism[5–7].

A second set of materials under extensive research are $ABO_3$ perovskite oxides in which multiferroicity is chemically engineered by placing ferroelectrically and magnetically active cations in the A- and B-site, respectively. Most promising systems are $BiMO_3$ compounds, such as $BiCoO_3$ and $BiFeO_3$, that are both ferroelectric and antiferromagnetic at room temperature, and for which magnetoelectric coupling has been demonstrated[8,9]. Indeed, $BiFeO_3$ stands out as the most topical multiferroic. This oxide has a rhombohedral distorted structure ($R3c$ space group) that results in spontaneous polarization exceeding $100 \mu C cm^{-2}$, and high transition temperature of $\sim 1,100 K$. Besides, it is a G-type antiferromagnet with an ordering temperature of $\sim 640 K$. Crystal magnetoelectric coupling exists and gives place to a long-range incommensurate cycloidal superstructure with the spins rotating within a plane containing the electrical polarization[10]. Spin-canting and weak ferromagnetism appear, this time as a result of the coupling between magnetism and the oxygen octahedral tilting, if the spin cycloid is destroyed[11,12]. This has been accomplished by strain engineering in thin films (for example, $M_S \approx 0.6 emu g^{-1}$ for 200 nm thick films)[13] and also by chemical modification (for example, $M_S \approx 1.2 emu g^{-1}$ for $Bi_{0.7}Ba_{0.3}FeO_3$ ceramic materials)[14].

Spontaneous polarization and magnetization easy axis are orthogonal in $BiFeO_3$, and the ferroelectric/ferroelastic and antiferromagnetic domain configurations are coupled, which enables the electrical drive of magnetic domains[15,16]. This effect has been used to achieve the electrical control of the exchange bias of a ferromagnetic layer deposited on top of the antiferromagnetic multiferroic by means of their exchange interaction[17,18]. Research on multiferroic heterostructures is being carried out in parallel to that on single-phase materials, and a wide range of magnetoelectric effects have been demonstrated[19–21].

Although $BiFeO_3$ is still at the cutting edge of research[22], latest efforts have concentrated in alternative perovskite systems like solid solutions between ferroelectric phases and relax or

compounds containing magnetically active cations, which show weak ferromagnetism in coexistence with a range of polar states, and different magnetoelectric coupling phenomena[23,24]. The underlying mechanisms are not clear, but spin clustering is thought to be responsible for the ferromagnetic component in the otherwise expected paramagnetic state, while elastic interactions between the proposed magnetic nanoregions and the ferroelectric matrix would cause the magnetoelectric effects. Partially reversible ferroelectric domain reorientations under magnetic fields have been recently described for $Pb(Zr_{0.53}Ti_{0.47})O_3$–$Pb(Fe_{1/2}Ta_{1/2})O_3$ at room temperature[25]. An increasing attention is also being paid to multiferroic Bi-containing layered perovskites, after similar effects were reported[26].

Perovskite solid solutions are also the basis of a different, not well explored approach. A strong phase-change magnetoelectric response has been anticipated by a first-principles investigation of phases in the $BiFeO_3$–$BiCoO_3$ perovskite binary system[27], associated with the presence of morphotropic phase boundary (MPB) between multiferroic polymorphs of rhombohedral $R3c$ and tetragonal $P4mm$ symmetries. Mechanism would be the discrete rotation of the magnetization easy axis following that of the spontaneous polarization under the electric field, linked to the electric field-induced transformation from one multiferroic polymorph to the other. Large phase-change electromechanical responses have been widely reported for perovskite systems with ferroelectric MPBs, like $Pb(Zn_{1/3}Nb_{2/3})O_3$–$PbTiO_3$[28]. Therefore, analogous magnetoelectric responses might be a general property of multiferroic MPBs, and a novel approach for room-temperature magnetoelectricity.

A recent experimental study reported the existence of a monoclinic $Cm$ phase between the rhombohedral and tetragonal polymorphs in the $BiFeO_3$–$BiCoO_3$ system, and polarization rotation with temperature and composition within this phase[29]. This phenomenology resembles that one described for ferroelectric MPBs in high-sensitivity piezoelectrics like $Pb(Zr,Ti)O_3$, which is directly associated with their very high piezoelectric response[30,31]. The possibility of obtaining an enhanced magnetoelectric response at multiferroic analogues showing lattice transverse softening was indicated by a recent theoretical study[32].

Here, we report a novel perovskite oxide chemically designed to present a multiferroic MPB that does show distinctive room-temperature magnetoelectric responses by means of these two mechanisms: field-driven phase changes and transverse lattice softening. The new material is a solid solution belonging to the $BiFeO_3$–$BiMnO_3$–$PbTiO_3$ ternary system. $BiFeO_3$–$PbTiO_3$ has been extensively studied as an alternative to $BiFeO_3$–$BiCoO_3$, which also shows a multiferroic phase boundary[33–35], but that can be obtained by mechanosynthesis[36]. A distinctive reversible, phase-change electromechanical response has been reported for this system[37], along with enhanced permittivity and piezoelectric coefficients across the MPB[38]. However, both polymorphic phases are antiferromagnetic, yet a weak ferromagnetic component appears at a second magnetic transition within the $BiFeO_3$-rich phase[38,39]. Recently, we described a full line of MPB compositions within the phase diagram of the $BiFeO_3$–$BiMnO_3$–$PbTiO_3$ ternary system[40]. Material here reported of composition $Bi_{0.68}Pb_{0.32}Fe_{0.655}Mn_{0.025}Ti_{0.32}O_3$ ($0.655BiFeO_3$–$0.025BiMnO_3$–$0.32PbTiO_3$) lies within this line, and shows both enhanced ferroelectricity and weak ferromagnetism for distinctive room-temperature magnetoelectric responses.

## Results

**Crystal structure and phase coexistence.** A first critical issue was determining the actual phase coexistence at the MPB, namely discriminating whether the polymorphic phase in coexistence

with the tetragonal component was rhombohedral or monoclinic, for lattice transverse softening is characteristic of the latter symmetry. This was done by using high-resolution X-ray diffraction (XRD). Chemically homogenous nanocrystalline powders obtained by mechanosynthesis, thermally treated at 950 °C for 12 h to increase crystallinity were characterized. A typical XRD pattern is shown in Fig. 1a. Phase coexistence between a tetragonal polymorph and a second one that might be either rhombohedral or monoclinic, in about equal amounts, is clearly observed.

The diffraction profile was then analyzed by the Rietveld method using JANA package[41]. The tetragonal $P4mm$ space group of $PbTiO_3$ along with the rhombohedral $R3c$ of $BiFeO_3$ or monoclinic $Cc$ symmetry (among others) were selected as structural models. The best fits between observed and calculated profiles obtained by the Rietveld refinements were compared with choose among the models. The initial lattice parameters and atom coordinates for the phases involved in the refinements were taken from previous reports on the $BiFeO_3$–$PbTiO_3$ binary system with composition at the MPB[34,35]. This is especially important for the

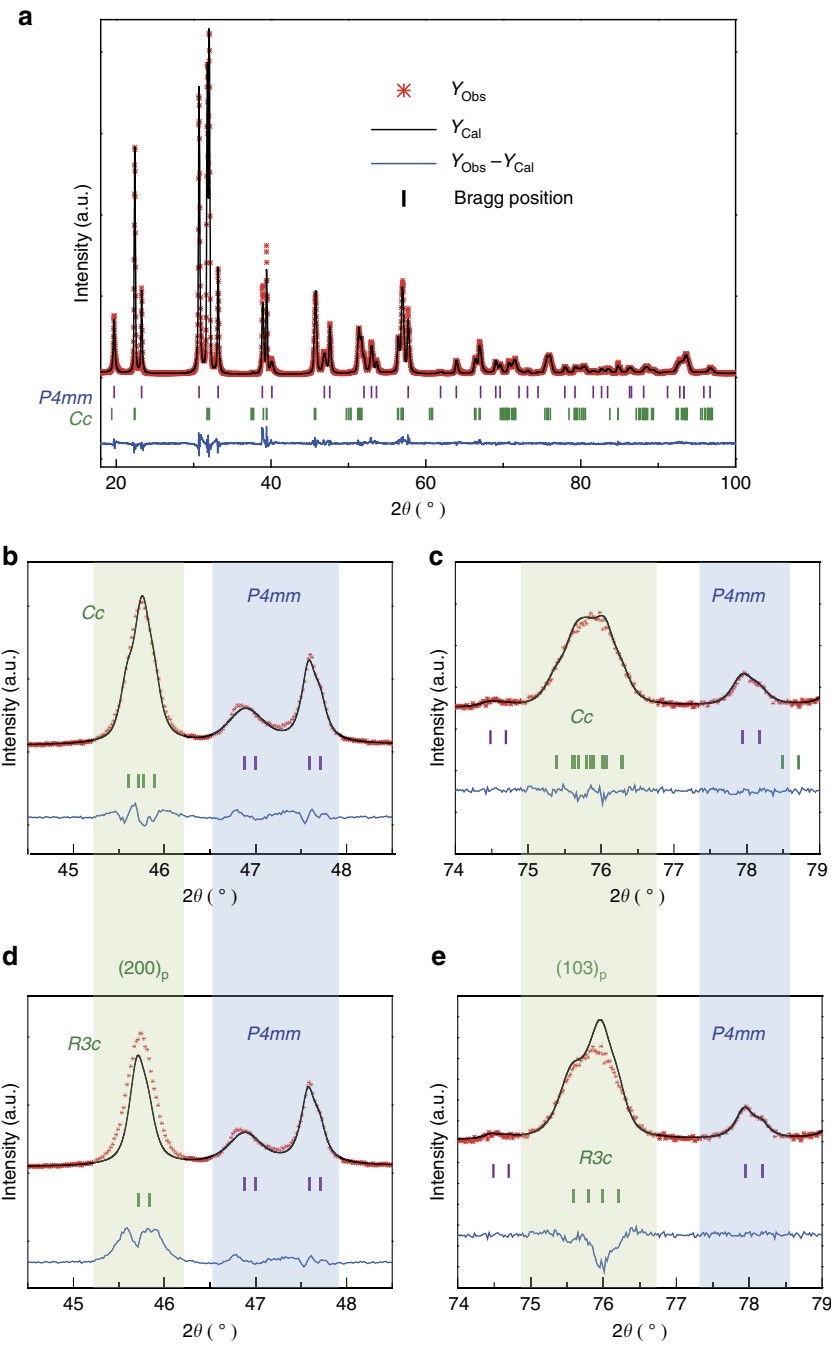

**Figure 1 | Crystal structure. (a)** High-resolution XRD pattern and Rietveld refinement for the $Bi_{0.68}Pb_{0.32}Fe_{0.655}Mn_{0.025}Ti_{0.32}O_3$ powder sample, at room temperature, indicating phase coexistence between polymorphs with monoclinic $Cc$ and tetragonal $P4mm$ symmetries. $Y_{Obs}$ and $Y_{Cal}$ refer to the measured and calculated intensities, respectively. The refined profiles using the (**b,c**) $Cc$ and (**d,e**) $R3c$ symmetries across the $(200)_p$ and $(103)_p$ reflections (referring to pseudocubic indices) are shown to illustrate the better agreement of the $Cc$ phase.

tetragonal polymorph, for it shows a huge tetragonal distortion of $c/a \sim 1.18$, which is considerably higher than that of $PbTiO_3$.

A major problem in the application of the Rietveld method to this system arises from the fits of the peaks at high angles, for which the complexity of the profile resulting from the polymorphic phase coexistence is a handicap. This is further complicated by the crystal stresses present due to the large difference in transition strain between the two coexisting polymorphs (anisotropy factors due to ferroelectricity need to also be considered). This requires the implementation of anisotropic strain broadening models, such as that proposed by Stephens[42]. At final stages, composition was released, and refinements confirmed the nominal composition for the two polymorphic phases as previously described (details of the refinements are given in Supplementary Note 1). The refined structural parameters for the $R3c$ and $Cc$ phases, using both mixture models involving coexistence with $P4mm$ polymorph, are given in Table 1 (refined position coordinates, thermal parameters and so on, are given in Supplementary Table 1).

It can be seen that the statistical parameters obtained, factors $R_{wp}$, $R_{exp}$ and the goodness-of-fit indicator, are better for the model involving the monoclinic $Cc$ space group. The best agreement for the latter phase is evident when one specifically analyses the refined profiles across the $(200)_p$ and $(103)_p$ reflections (referring to pseudocubic indices), shown in Fig. 1b,c for the $Cc$ and $R3c$ symmetries, respectively. It is clear that the fit using $R3c$ symmetry is not satisfactory, for example, the anomalous peak broadening of the $(200)_p$ reflection is not consistent with its singlet nature (and neither it is that of the $(103)_p$ considering it as a doublet), as expected for the $R3c$ space group.

To confirm the actual existence of a monoclinic $Cc$ polymorphic phase, the study was also carried out in another composition at the pseudo-rhombohedral side of the MPB, right next to that under study, but for which the tetragonal phase is not present (that is, outside the MPB region). This composition is $Bi_{0.75}Pb_{0.25}Fe_{0.7}Mn_{0.05}Ti_{0.25}O_3$, for which only one polymorph can be observed in the XRD pattern, as shown in Supplementary Fig. 1, thus avoiding the complexity of refining profiles exhibiting phase coexistence. The profile was then refined for both symmetries $R3c$ and $Cc$, among others. The refined structural and statistical parameters are summarized in Supplementary Table 2. Once again, statistical parameters and goodness-of-fit indicator are much better for the monoclinic $Cc$, in particular $R_{Bragg}$.

**Ferroelectricity and phase-change phenomena.** Fully dense materials with tailored microstructures and no traces of chemical inhomogeneities (checked by scanning electron microscopy and energy dispersive XRD) were processed by hot-pressing at moderate temperatures of the nanocrystalline powder obtained by mechanosynthesis. This is a requirement for a sound ferroelectric

characterization, and also for applicability, which is not easily achieved in the case of low-tolerance factor perovskites, such as $BiFeO_3$- and $BiMnO_3$-based materials. Specifically, incipient perovskite decomposition during sintering can result in increasing levels of A-site and oxygen vacancies within the structure, and also of Fe and Mn mixed-valence states. Actually, this is a characteristic of these materials that always show non-negligible electron hopping-type conductivity associated with the presence of such defects, even though the microstructural characterization did not show any evidence of phase decomposition or grain boundary minor secondary phases in the ceramics.

The temperature and frequency dependences of permittivity showed a Maxwell–Wagner type relaxation above room temperature. This is a distinctive step-like increase of permittivity at a temperature that shifts with frequency, from about 475 K at 1 kHz to 650 K at 1 MHz in the case of $Bi_{0.68}Pb_{0.32}Fe_{0.655}Mn_{0.025}Ti_{0.32}O_3$ (Supplementary Fig. 2). Maxwell–Wagner relaxations are associated with the existence of significant, but inhomogeneous electrical conductivity in the material, a phenomenon widely observed in ceramic materials as a result of the presence of highly resistive grain boundaries[38]. Conductivity at moderate temperatures in this specific case is associated with electron hopping between mixed-valence state transition metals, $Fe^{3+}/Fe^{2+}$ and $Mn^{3+}/Mn^{4+}$, as indicated by the obtained activation energy of 0.4 eV (see inset in Fig. 2a). Note that at high temperatures above 550 K the activation energy increases up to 1.1 eV, a value typical of mobile oxygen vacancies, which would be, not-surprisingly, the mechanism of the high-temperature conductivity. It is important to highlight that the onset of the Maxwell–Wagner polarization, and thus of electrical conduction takes place at 450 K for $Bi_{0.68}Pb_{0.32}Fe_{0.655}Mn_{0.025}Ti_{0.32}O_3$. Tailored ceramic materials show typical room-temperature resistivity values of $10^9 \, \Omega \, cm$, comparable to those of $BiFeO_3$ and other ferroelectrics.

This tailored conductivity enabled the ferroelectric characterization. A summary of the ferroelectric properties is shown in Fig. 2. The temperature dependence of the dielectric permittivity at 1 MHz, measured during a heating/cooling cycle is shown in Fig. 2a. The permittivity maximum associated with the ferroelectric to paraelectric transition, $T_C$, is clearly observed. The transition shows a significant thermal hysteresis during the measuring cycle, with $T_C$ of 900 and 880 K on heating and cooling, respectively. These values are close to those reported for the $0.675BiFeO_3$–$0.325PbTiO_3$ binary system, c.a. 915 and 880 K, and about 200 K lower than those for $BiFeO_3$ (ref. 38). The large hysteresis is consistent with results obtained in the binary system, for which an anomalous large $\Delta T_C$ was also found at the MPB, and associated with a different sequence of polymorphic transitions on heating and cooling, likely from the tetragonal phase on heating and to the monoclinic one on cooling[38].

A low-temperature ferroelectric state exists then for this material, as confirmed by P-E hysteresis loop measurements. Typical ferroelectric hysteresis and current density curves after

**Table 1 | Refined structural parameters and agreement factors.**

| Space group | R3c | P4mm | Cc | P4mm |
|---|---|---|---|---|
| Statistical parameters | $R_{wp} = 10.86$; $R_p = 7.97$; $R_{exp} = 3.80$; GOF = 2.86 | | $R_{wp} = 8.29$; $R_p = 6.15$; $R_{exp} = 3.79$; GOF = 2.19 | |
| Lattice parameters (Å) | $a = 5.5883$ | $a = 3.8185$ | $a = 9.7914$ | |
| | $c = 13.8438$ | $c = 4.4943$ | $b = 5.5829$ | $a = 3.8188$ |
| | | | $c = 5.6240$ | $c = 4.4946$ |
| | | | $\beta = 125.69°$ | |
| Phase fraction | 0.54($R3c$)/0.46($P4mm$) | | 0.56($Cc$)/0.44($P4mm$) | |

Results of the Rietveld refinements of the $Bi_{0.68}Pb_{0.32}Fe_{0.655}Mn_{0.025}Ti_{0.32}O_3$ powder composition, using mixture models with coexistence of rhombohedral $R3c$ and tetragonal $P4mm$, and of monoclinic $Cc$ and tetragonal $P4mm$. Reliability factors (R-factors) $R_{wp}$, $R_p$, $R_{exp}$ and goodness-of-fit indicator (GOF) are given.

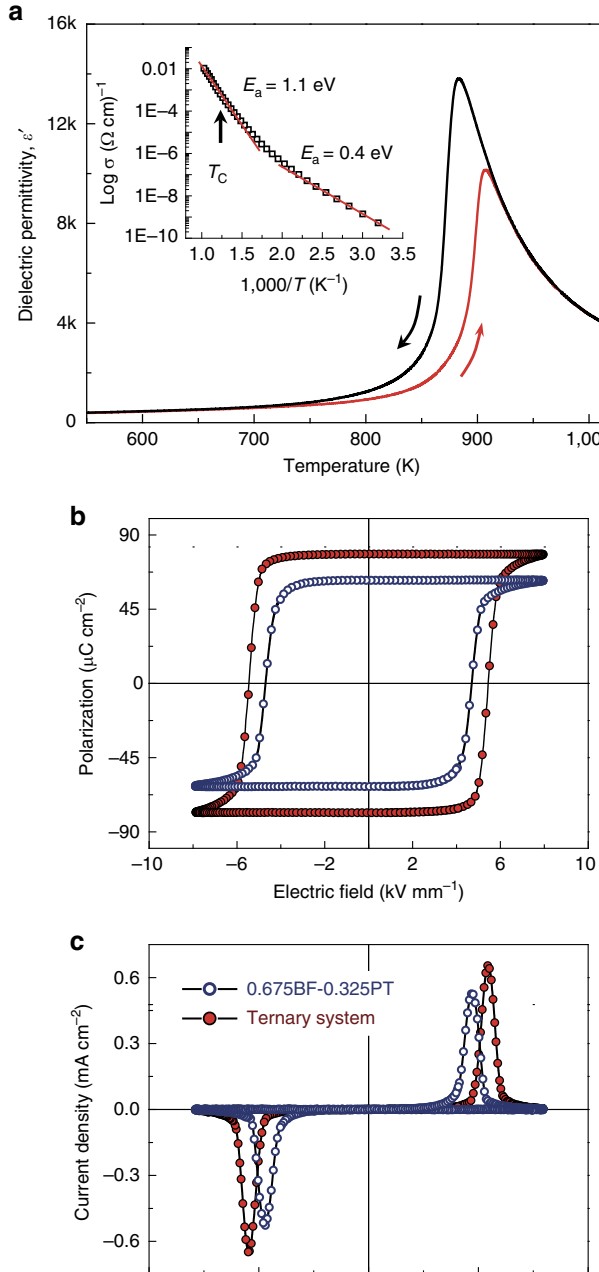

**Figure 2 | Ferroelectric properties.** (**a**) Temperature dependence of the relative permittivity at 1 MHz of the $Bi_{0.68}Pb_{0.32}Fe_{0.655}Mn_{0.025}Ti_{0.32}O_3$ ceramic, showing significant thermal hysteresis on a heating and cooling cycle (Inset: dc conductivity ($\sigma$) vs inverse absolute temperature ($T$). Electrical ordering temperature ($T_C$) and activation energy ($E_a$) values are indicated. (**b**) P-E hysteresis loops and (**c**) current density curves for the ceramic under study (ternary system) and those for the MPB composition of the binary system (0.675$BiFeO_3$–0.325$PbTiO_3$) for comparison.

compensation are shown in Fig. 2b,c, that is, after subtracting linear polarization and conduction contributions (Supplementary Fig. 3). Note the high squareness of the loop that is more typical of a single crystal than of a ceramic material. It is highly remarkable the very high remnant polarization achieved, $P_r \approx 78\,\mu C\,cm^{-2}$, the largest value ever reported for a ferroelectric ceramic so far, and the comparatively low-coercive field $E_c$ of 5.4 kV mm$^{-1}$ ($E_c$ above 8 kV mm$^{-1}$ has been reported for

rhombohedral $BiFeO_3$–$PbTiO_3$[38]. These figures favourably compare with those for the binary system at its MPB: $P_r \approx 62\,\mu C\,cm^{-2}$ and $E_c \approx 4.7\,kV\,mm^{-1}$. Loops for these materials could only be obtained after quenching from temperatures above the ferroelectric transition. This is an effective means of releasing the ferroelectric domain walls, initially clamped by oriented $Fe_{Fe}'$-$V_O^{\bullet\bullet}$ dipolar defects[37]. Note also that fully saturated loops could only be obtained at 250 K, for dielectric breakdown occurred at room temperature under electric driving above 5 kV mm$^{-1}$.

**Magnetization and spin-reorientation phenomena.** $Bi_{0.68}Pb_{0.32}$ $Fe_{0.655}Mn_{0.025}Ti_{0.32}O_3$ not only showed enhanced ferroelectricity as compared with the binary system, but also enhanced spin-canted ferromagnetism. Magnetization was measured during conventional zero-field-cooling (ZFC)/field-cooling (FC) cycles under a dc magnetic field of 500 Oe. The temperature dependence of the molar magnetic susceptibility (M/H ratio) is shown in Fig. 3a for the ceramic material whose ferroelectric properties have just been presented. Note that the ZFC and FC curves do not still overlap at 400 K and that the system maintains significant magnetization at this temperature, which indicates the anti-ferromagnetic ordering temperature to be above our measuring range. Indeed, Néel temperature ($T_N$) above 400 K has been reported for compositions at the MPB of the $BiFeO_3$–$PbTiO_3$ binary system (c.a. 480 K)[33].

The key feature is the magnetic anomaly at ~360 K. This anomaly was firstly described for MPB compositions of the $BiFeO_3$–$PbTiO_3$ binary system, and associated with a spin-reorientation (SRO) phenomenon within the antiferromagnetic state of the monoclinic $Cc$ phase[39]. We have recently shown that this spin reorientation takes place at ~380 K for 0.7$BiFeO_3$–0.3$PbTiO_3$, and shifts towards lower temperatures upon addition of $BiMnO_3$, along the line of MPBs in the ternary system[40]. The SRO is accompanied by the appearance of a large divergence between ZFC and FC curves on cooling, which indicates a significant increase of the net magnetic moment resulting from spin-canting. Indeed, magnetization loops recorded at 300 K show a remnant value of ~0.02 emu g$^{-1}$, as shown in Fig. 3b. This weak ferromagnetic moment, although small when compared with that of other $BiFeO_3$-based compositions like $Bi_{0.7}Ba_{0.3}FeO_3$[14], takes place within a monoclinic phase allowing lattice transverse softening. Enhanced magnetoelectric coefficients can then be expected on approaching the instability of the multiferroic state[32]. The divergence between the ZFC and FC curves below the SRO temperature increases with the addition of $BiMnO_3$ along the line of MPBs (see loops in Fig. 3b), at the same time that the SRO shifts below room-temperature for high $BiMnO_3$ contents (Supplementary Fig. 4). $Bi_{0.68}Pb_{0.32}Fe_{0.655}$ $Mn_{0.025}Ti_{0.32}O_3$ is the composition along the line of MPBs that has the largest ferromagnetic component, while maintaining the SRO above room temperature.

Finally, note the low-temperature anomaly taking place at ~100 K, which is hard to distinguish in the ZFC/FC curves but easily noticeable in their derivatives (see inset of Fig. 3a). This is thought to be the Néel temperature of the tetragonal polymorph, as previously reported for the binary system at ~200 K (ref. 33). The coexistence of two polymorphs with very different room-temperature magnetism, a monoclinic antiferromagnet with enhanced canted ferromagnetism and a tetragonal paramagnet was required for obtaining the targeted phase-change magnetoelectric response.

**Magnetoelectric coupling.** $Bi_{0.68}Pb_{0.32}Fe_{0.655}Mn_{0.025}Ti_{0.32}O_3$ is thus a perovskite oxide placed at a discontinuous multiferroic

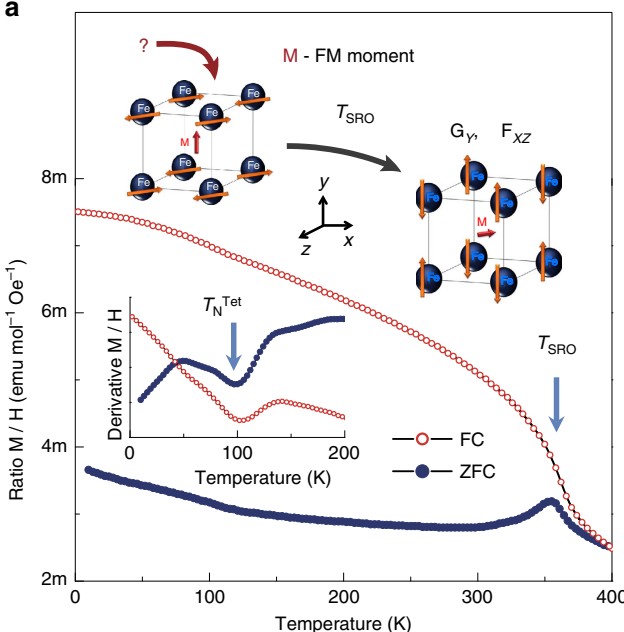

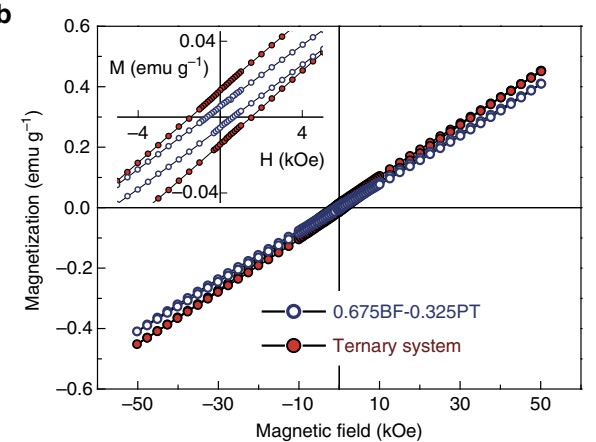

**Figure 3 | Magnetic properties.** (**a**) Zero-field-cooled (ZFC)/Field-cooled (FC) magnetization curves for the $Bi_{0.68}Pb_{0.32}Fe_{0.655}Mn_{0.025}Ti_{0.32}O_3$ ceramic under a dc magnetic field of 500 Oe. The magnetic anomaly at 360 K is associated with a spin-reorientation phenomenon ($T_{SRO}$) within the $Cc$ phase (Inset shows a schematic description of the proposed magnetic structure involved in the SRO in terms of ferromagnetic F and antiferromagnetic G components[39], M stands for the ferromagnetic (FM) moment. Derivative curves highlights the Néel temperature ($T_N^{Tet}$) at ∼100 K of the $P4mm$ phase. (**b**) Isothermal magnetization loop recorded at 300 K for the ceramic under study (ternary system) and that for the MPB composition of the binary system (0.675BiFeO₃–0.325PbTiO₃) for comparison.

phase boundary between a tetragonal $P4mm$ phase and a monoclinic $Cc$ phase with enhanced canted ferromagnetism, which also shows very high remnant polarization. It is therefore an ideal system to explore the occurrence of large magnetoelectric effects at this type of instabilities of the multiferroic state. The feasibility of having a phase-change response was explored by comparing the magnetic response of unpoled and poled ceramics, for the percentages of coexisting polymorphs are known to change upon poling as checked by XRD (Supplementary Fig. 5; Supplementary Note 2). This was accomplished by applying a very-low frequency (0.01 Hz) sine wave of $8\,kV\,mm^{-1}$ amplitude, and by removing the field just before completing the

ferroelectric hysteresis loop. A longitudinal piezoelectric charge coefficient $d_{33}$ of $62\,pC\,N^{-1}$ was obtained as measured with a Berlincourt meter.

The ZFC/FC magnetization curves at 500 Oe for the same ceramic sample before and after the poling process are shown in Fig. 4a. Main features like the SRO anomaly within the monoclinic phase and the low-temperature step associated with the Néel temperature of the tetragonal component are preserved, but there is a distinctive decrease of the divergence between the ZFC and FC curves below the SRO anomaly after poling, indicating a decrease of the net magnetic moment in the poled material. This suggests a partial transformation of the monoclinic into tetragonal phase, which is consistent with the increase of the height of the low-temperature step. Indeed, the magnetization loop for the poled material shows less than half the remnant value (below $0.01\,emu\,g^{-1}$) than the unpoled one, as shown in Fig. 4b.

Also, the linear magnetoelectric response at room temperature was studied for the poled ceramic material as a function of magnetization. This was carried out by measuring the magneto-electric voltage developed under the application of an alternate low-magnetic field, while the samples were subjected to static, increasingly high-magnetization fields. A longitudinal-transverse configuration; polarization perpendicular to magnetic fields, was selected. The results are shown in Fig. 4c for opposite directions of polarization. The change of sign in the slope is a direct proof of the magnetoelectric origin of the response, which is well above the background signal of the equipment that originates from magnetic induction. A magnetoelectric voltage coefficient $\alpha_{31}$ of ∼$0.5\,mV\,cm^{-1}\,Oe^{-1}$ was calculated after subtracting the background from the voltage amplitude at maximum dc field. It is worth remarking that this response must originate only from the monoclinic component, decreased in the poled state, for the tetragonal phase is in the paramagnetic state at room temperature.

## Discussion

There are a few issues that deserve further discussion, such the actual existence of a monoclinic $Cc$ polymorphic phase. Indeed, a controversy persists regarding the presence of this $Cc$ polymorph at the MPB of the binary system. The crystal structures reported for BiFeO₃–PbTiO₃ by different groups differ and seem to depend on the synthesis/processing conditions and also on the thermal history[43]. The same composition 0.7BiFeO₃–0.3PbTiO₃ has been claimed to be a mixture of tetragonal $P4mm$ and either rhombohedral $R3c$ (ref. 44) or monoclinic $Cc$ (ref. 35) structures. However, there are a number of distinctive magnetic and electrical features of BiFeO₃–PbTiO₃ that seem to be characteristic of the monoclinic symmetry, such as the magnetic SRO transition[39], and the existence of transverse lattice softening[38]. In the case of the ternary system, we have found phase coexistence between $P4mm$ and $Cc$ polymorphs for a composition not far from 0.7BiFeO₃–0.3PbTiO₃. Whether this was already monoclinic or it transformed from rhombohedral into monoclinic by BiMnO₃ addition is beyond the scope of this article.

We have also found strong thermal history effects in the phase coexistence. The percentages of the coexisting $Cc$ and $P4mm$ components significantly change for the same powdered sample whether the material is quenched or slowly cooled from temperatures above the ferroelectric transition. This has also been observed for BiFeO₃–PbTiO₃, and nicely illustrates how close polymorphic phases are in the energy space, which is a key feature to enable phase-change phenomena[31]. The large thermal hysteresis revealed by the dielectric characterization of ceramics in the ferroelectric transition is another manifestation of the

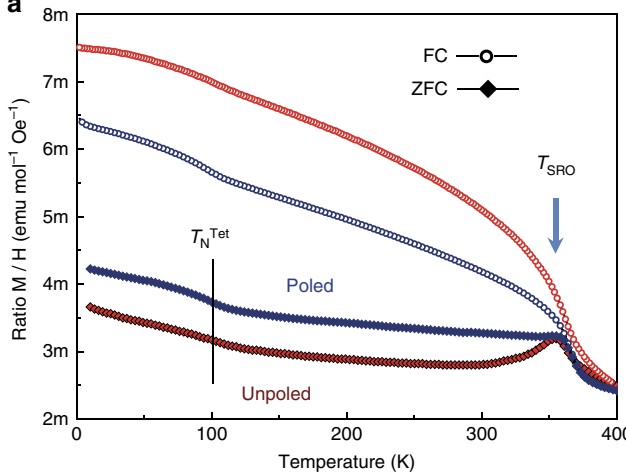

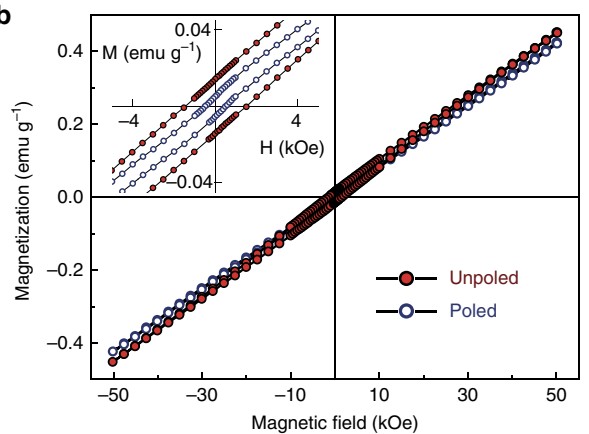

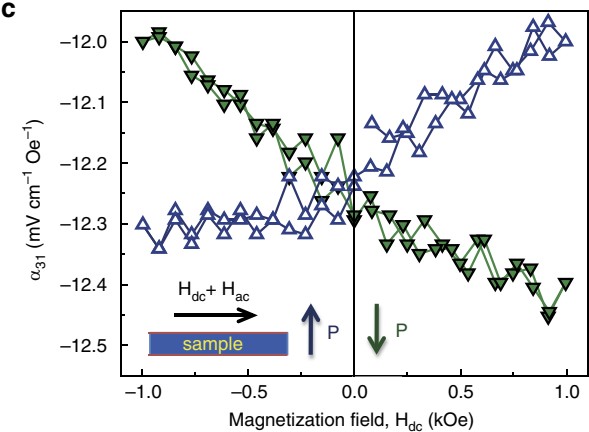

**Figure 4 | Magnetoelectric coupling.** (**a**) Zero-field-cooled (ZFC)/Field-cooled (FC) magnetization curves at 500 Oe and (**b**) isothermal magnetization loops at 300 K, for the unpoled and poled $Bi_{0.68}Pb_{0.32}Fe_{0.655}$ $Mn_{0.025}Ti_{0.32}O_3$ ceramic. Néel temperature ($T_N^{Tet}$) of the $P4mm$ phase and magnetic anomaly associated with the spin-reorientation phenomenon ($T_{SRO}$) within the $Cc$ phase are indicated. A distinctive magnetoelectric response is anticipated, associated with the observed electric field-induced phase change between $Cc$ and $P4mm$ phases. (**c**) Linear magnetoelectric response was also found, as indicated by the magnetoelectric voltage coefficient ($\alpha_{31}$) as a function of a static magnetization field, $H_{dc}$ (open and closed symbols are for polarization pointing in antiparallel directions. The measuring configuration is illustrated in the inset).

energy degeneracy. This was previously associated with a different sequence of polymorphic transition on heating and cooling, specifically from the $P4mm$ phase on heating, and to the $Cc$ one on cooling, as suggested by the evolution of the Curie temperatures along the $BiFeO_3$–$PbTiO_3$ system[38]. In this work, the temperature evolution of the XRD profiles was studied for $Bi_{0.68}Pb_{0.32}Fe_{0.655}Mn_{0.025}Ti_{0.32}O_3$ powders, and well differentiated phase transition temperatures were obtained for the monoclinic and tetragonal polymorphs (Supplementary Fig. 6). Indeed, this was lower for the former phase in good agreement with the thermal hysteresis described from the electrical measurements. Also an effect of stress on the phase coexistence, ceramic samples show an enhanced percentage of the monoclinic phase as compared with powders (Supplementary Fig. 5).

The latter phenomenology would be the stress-induced phase-change phenomena, but electric field-induced phase-change effects are also indicated by ferroelectric hysteresis loops. Indeed, the extremely high electrical polarization of $78\,\mu C\,cm^{-2}$ found for $Bi_{0.68}Pb_{0.32}Fe_{0.655}Mn_{0.025}Ti_{0.32}O_3$ also merits an in-depth discussion. Note that this polarization figure is for a polycrystalline ceramic, and so, the crystal spontaneous polarization must be significantly higher. Its evaluation would require knowledge of the actual switching mechanism, usually of whether nucleation and growth of non-180° ferroelectric domains contribute along with 180° ones to the switching. It has been demonstrated that 90° domains of ultrahigh tetragonal distortion $BiFeO_3$–$PbTiO_3$ are immobile[45], which agrees with the high squareness of the loops. Some polarization relaxation always takes place when non-180° contributions exist due to mechanical stresses built during the process, which is not observed here within the loop time scale. In this case, and accepting that only 180° domain reversals contribute to ferroelectric switching, simple geometry/crystallographic arguments state that the spontaneous polarization would be twice the saturation polarization value. This would mean a crystal polarization above $150\,\mu C\,cm^{-2}$, even larger than that of $BiFeO_3$ (ref. 46) and $BiFeO_3$–$PbTiO_3$ (ref. 37).

Although theoretical calculations have predicted giant spontaneous polarization up to such numbers for epitaxially-strained tetragonal-like $BiFeO_3$ thin films[47], and experimental evidence of $\sim 130\,\mu C\,cm^{-2}$ was reported[48], this was for materials with a huge $c/a$ ratio of 1.27. This is actually much higher than tetragonality of our sample, $c/a \sim 1.18$, and indeed an estimation of the spontaneous polarization from the ionic positions provided by the Rietveld analysis gives values of 98 and $123\,\mu C\,cm^{-2}$ for the monoclinic and tetragonal polymorphs, respectively (see Supplementary Table 3 and Supplementary Note 3 for details). Therefore, an additional mechanism to domain wall nucleation and movement must be active in this system, and polarization figures are consistent with it being phase changes between the two coexisting polymorphs under the electric field, which would take place by nucleation and movement of anti-phase boundaries. In this case, and accepting now that unfavourably aligned domains also contribute to ferroelectric switching through the phase-change mechanism, the spontaneous polarization would be about $110\,\mu C\,cm^{-2}$, in a good agreement with the estimation from structural refinements and also consistent with the irreversible field-driven phase change upon poling (as seen in Supplementary Fig. 5). It is worth to recall that a distinctive phase-change electromechanical response has been reported for $BiFeO_3$–$PbTiO_3$ (ref. 37).

This electric field-induced phase-change takes place in a system with magnetic properties, specifically between a monoclinic phase with spin-canted antiferromagnetic ordering, and a tetragonal phase with $T_N$ below room temperature. Therefore, a distinctive magnetoelectric response must exist associated with the electric field-induced phase changes. Indeed,

$Bi_{0.68}Pb_{0.32}Fe_{0.655}Mn_{0.025}Ti_{0.32}O_3$ has been chemically designed to show a multiferroic MPB with enhanced canted ferromagnetism, and it is thus especially suited to test the actual existence of a phase-change magnetoelectric response. This has been again unambiguously proved by the changes in the magnetism of a ceramic upon poling, which involves a partial monoclinic to tetragonal phase transformation. Accordingly, a reversible magnetization change can be anticipated during the unipolar electrical driving (parallel to the poling) of a poled ceramic, associated with the expected, yet slow partial phase-change relaxation after removing the electric field.

The previous phenomenology would be a high-field magneto-electric response, but a low-field linear magnetoelectric one has also been found. Phase changes are not expected at low-driving fields, so this response must be associated with the linear magnetoelectric effect, and specifically with that of the mono-clinic component, for the tetragonal phase is paramagnetic at room temperature. Weak ferromagnetism like that described after the SRO is necessary for the linear magnetoelectric effect to happen[12]. A schematic drawing of the recently reported G-type antiferromagnetic arrangements of the $Fe^{3+}$ spins in the perovskite unit cell and the superimposed spin-canting originating the ferromagnetic moment is shown in the inset of Fig. 3a (ref. 39). The canting would be analogous to that described for $BiFeO_3$, and that is known to be intrinsically linked with the oxygen octahedral tilting[11,12]. A $(G_y,F_{xz})$ magnetic structure was proposed above the SRO[39], which nicely obeys the universal law stating that the ferromagnetic moment must be perpendicular to both the antiferromagnetic vector and the octahedral rotation axis[49]. However, the analysis is not straightforward for the $(G_{XZ},F_Y)$ structure proposed below the SRO, for the former law is not fulfilled. Though the calculation of the actual octahedral rotation axis, which requires the accurate determination of the antiferrodistortive motions, is beyond the scope of this article, this axis must be of the [uuv] type (Supplementary Note 4) that is not perpendicular to the proposed direction of the ferromagnetic moment.

The voltage magnetoelectric coefficient $\alpha_{31}$ $\sim 0.5\,mV\,cm^{-1}\,Oe^{-1}$ obtained is comparable to the experimental one reported for $BiFeO_3$ after the destruction of the magnetic cycloid under very high magnetic fields ($\sim 1\,mV\,cm^{-1}\,Oe^{-1}$), and also to those obtained from first-principles[12] (calculated from the charge coefficients by assuming a relative permittivity of 50)[38]. They are also comparable with best values reported at room temperature for other perovskite single-phase materials[50]. Note that higher coefficients should be obtained by maximizing the percentage of monoclinic phase, yet a compromise must be reached if phase-change responses are also targeted.

Although further research is necessary to fully explore the potential of this approach, we believe that results here presented demonstrate that both linear magnetoelectric effect and large phase-change magnetoelectric response can be obtained at multiferroic phase boundaries. The novel perovskite oxide here reported, chemically designed to show this type of multiferroic instability and enhanced canted ferromagnetism, shows the targeted phase-change phenomena, and also linear magneto-electric effect, all at room temperature. This phenomenology is analogous to that previously described for $BiFeO_3$ epitaxial films under high compressive stress, in which a multiferroic MPB is strain engineered[51]. Indeed, electric field-induced rhombohedral (actually monoclinic $Cc$) to tetragonal phase transformation has been demonstrated with a distinctive effect on magnetism[52]. We report here analogous effects in a single-phase material, without the need of epitaxial thin film technology and strain engineering. This enables the possibility of using ceramic materials, either bulk or thin films on Pt/Si-based substrates, which is clearly advantageous for their potential application. Note for instance that commercial FeRAMs are based on polycrystalline PZT films on Pt/Si-based substrates. The use of the new material for memory applications is not straightforward, for it requires a procedure to revert the phase change. This has been achieved in mixed phase $BiFeO_3$ epitaxial films by applying a moderate (below the coercive one) opposite electric field[52]. Magnetoelectric phase-change phenomena have also been anticipated in other perovskite systems, such as Nd substituted $BiFeO_3$, associated with a field-induced orthorhombic $Pnma$ to rhombohedral $R3c$ transition[53]. We reckon, therefore, this new material approach for room-temperature magnetoelectricity to be very promising, and potentially capable of enabling the range of magnetoelectric technologies that are eagerly awaiting a suitable material.

## Methods

**Synthesis of nanopowders and ceramic processing.** Mechanosynthesis was used to obtain chemically homogenous nanocrystalline powders of the novel perovskite phase. Details of the procedure and of the mechanisms taking place during the mechanical treatments can be found elsewhere[36,40]. Thermal treatments at 950 °C for 12 h followed by fast cooling (quenching) in air down to room temperature were carried out to increase the powder crystallinity for the XRD studies. Ceramic materials were directly processed from the nanocrystalline powders by hot-pressing at 950 °C for 1 h. A stringent control of the preparation conditions was required to obtain high-quality ceramic materials[38].

**Instrumentation and characterization.** X-ray diffraction patterns were collected on a two-axis diffractometer with a Rigaku RA-HF18 rotating anode generator (Cu $K_\alpha$ radiation). Electrical properties were measured on ceramic discs, on which Au electrodes were painted and annealed at 750 °C for 1 h. The temperature dependence of the dielectric permittivity was dynamically measured under heating/cooling cycles at 1.5 °C min$^{-1}$, with a HP4284A precision LCR Meter in the frequency range 20 Hz–1 MHz. Additional impedance spectroscopy was carried out in static conditions at temperature steps of 20 °C in the same frequency range. Ferroelectric hysteresis loops were obtained by current integration. High-voltage sine waves (0.1 Hz) were applied by the combination of a synthesizer/function generator (HP3325B) and a high-voltage amplifier (Trek Model 10/40A), while charge was measured with a homebuilt charge to voltage converter. Loops are presented after subtracting linear polarization and conduction contributions to the current curves, assuming a resistance and a capacitance in parallel.

Magnetic properties were characterized with a Quantum Design MPMS-XL5 SQUID magnetometer. The samples were cooled down to 2 K under zero field and then a magnetic field of 500 Oe was applied and maintained during a following heating (ZFC) and cooling (FC) cycle. Isothermal magnetization measurements M(H) between −50 kOe and +50 kOe were recorded at 300 K. The magnetoelectric response was measured using an experimental set-up made of a combination of two Helmholtz coils: a high power coil and a high frequency one, designed to independently provide a static increasingly high magnetic field up to 1 kOe (magnetization field), and an alternate magnetic field of 10 Oe at 10 kHz (the stimulus), respectively (Serviciencia S.L.). The out-put voltages developed as a response of the alternate magnetic field were monitored with a lock-in amplifier.

**Data availability.** The authors declare that all data supporting the findings of this study are included within the manuscript (and its Supplementary Information files) and are available from the corresponding author on request.

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

## Acknowledgements

This work was supported by Spanish MINECO projects (MAT2014-58816-R and MAT2011-23709). C.M.F.-P. thanks financial support of the Spanish FPI Programme (BES 2012-053017). Technical supports by Ms I. Martínez and Ms M.M. Antón, both at ICMM-CSIC, are also acknowledged.

## Author contributions

C.M.F.-P. and H.A. prepared the materials and performed most of the experiments, while A.C. and M.A. joined them in discussion and data analysis. J-M.K. and F.P. supported and discussed the structural studies, and O.P. the magnetic characterization. A.C. and H.A. designed and supervised the work. Finally, H.A. and M.A. co-wrote the manuscript with inputs from all authors.

## Additional information

**Competing financial interests:** The authors declare no competing financial interests.

