## [Peer review file · Nature Communications]

Reviewers' Comments:

Reviewer #1 (Remarks to the Author)

The authors present an experimental study aimed at obtaining a material switchable between two ferroelectric states ("T" and "R" or "Cc") that have different magnetic structures at room temperature, which would yield a robust "phase change" magnetoelectric effect at ambient conditions. This is certainly worth achieving, granting the interest of the work. Further, while my knowledge of the experimental aspects of these materials is limited, the work seems to be of a very high quality as regards the preparation and characterization of the samples. Hence, my opinion is quite positive. However, before making a final decision, I need the authors to help me dispel some doubts, and maybe make some revisions to their manuscript.

(1) The authors present indirect evidence that the phase-change magnetoelectric effect can be obtained, by investigating poled/unpoled 0 samples that are (zero-)field-cooled. While their story makes sense, one would have preferred to see a direct characterization of the field-driven phase change. In particular, it would be important to quantify which volume percentage it is possible to switch for experimentally attainable electric fields, as well as explicitly quantify the effect on the magnetic moment. The robustness and stability of the transformed structure, as required for some envisioned applications (memories), is another important point that I have missed in the present manuscript.

(2) "R" or "Cc" to "T" transformations have been explicitly demonstrated experimentally in thin films. Further, phase-change-like rotations of the canted magnetic moment (upon polarization switching between differently oriented domains) have can be obtained in BiFeO₃ films, and then characterized (and used) via exchange bias. These are room temperature effects. I think the authors should do a better job to explain why their strategy is superior to existing ones.

(3) I found the discussion on the linear magnetoelectric effect, associated to the rotation of the polarization in the "Cc" phase, interesting. Yet, at some parts of the manuscript, it was not clear to me whether the discussion was about this response or about the phase-change effect.

For example, in the paragraph that starts with "Furthermore, a linear magnetoelectric response has also been found, which must originate in the monoclinic component ..." the authors also write "The

transition takes place between two non- collinear antiferromagnetic states, wherein the spins change their magnetic easy axis altogether from the xz plane to be confined along the y axis...". Talking about a "transition" is clearly reminiscent of the phase-change effect. But then, in that case, the transition is between AFM and PM states, at least at room temperature. I find these and similar comments quite

confusing, and suggest the authors revise them.

(4) Why not study the magnetoelectric response of the unpoled samples which, if I understand things correctly, should be considerably larger?

(5) The magnetoelectric response of the "Cc" phase might bear similarities with the response of BiFeO₃ films grown on square (001)-oriented substrates, which are also monoclinic Cc. These have been investigated in detail experimentally (Ramesh and others after the famous 2003 Science paper) and theoretically (PRL 105, 037208 (2010)). It would be interesting if the authors could relate their current results to those works.

(6) Finally, it is not clear to me whether the presence of a "Cc" phase, rather than "R", is convenient in order to obtain a robust phase-change response. I would appreciate it if the authors could comment on that.

Reviewer #2 (Remarks to the Author)

The authors have designed, grown and characterized a perovskite solid solution that exhibits a multiferroic morphotropic phase boundary with room-temperature magnetoelectricity.

This concept is rather novel and of interest. As such, I believe it has the potential to be published in Nature Communications.

However, before being able to fully recommend this article for such publication, several important issues need to be addressed, in my mind:

- 1) there are some previously published works that appear to have been unfortunately overlooked by the authors, despite their relevance to the present study. For instance, Xu et al, *Advanced Functional Materials* 25, 552 (2015) also studied magnetoelectric effects in compositional area of some solid solutions associated with structural changes. Similarly, Zhao et al, *Nature Communications* 5, 4021 (2014) proposed the design of multiferroic materials having an electrical polarization and ferromagnetism near room-temperature. It is important to quote but also discuss/compare the results of these papers with those of the present manuscript, especially indicating the novelty of this latter manuscript.

- 2) I found the sentence on page 4 about BiFeO₃ (BFO) "Crystal magnetoelectric coupling exists...cancels out the macroscopic magnetization [10]" difficult to understand and easy to confuse readers. In particular one may wrongly believe, by reading this sentence, that it is the same coupling that results in the formation of a weak magnetization and of the magnetic cycloid, while in BFO, there are (at least) two different types of structural-magnetic interactions (both of DM types): one coupling oxygen octahedral tilting and magnetic moments and one coupling polarization and magnetic moments. The former one is responsible for the creation of the weak ferromagnetic moment (via spin-canting) and for the linear magneto-electric effect (please see Albrecht et al, *Physical Review B* 81, 140401 (R) (2010) and their reference 10). The latter is responsible for the formation of the magnetic cycloid (please see Rahmedov et al, *Physical Review Letters* 109, 037207 (2012)). The authors should rewrite their sentence to emphasize such facts, and thus to avoid any possible confusion.

- 3) Are the authors sure that no chemical decomposition occur in their sample? For instance, is it possible that the tetragonal phase arises in PbTiO₃-rich regions while the monoclinic phase occurs in BiFeO₃-rich regions? If such decomposition does occur, could it be that the PbTiO₃-rich regions feel a "negative" pressure coming from its surrounding, which will explain the large c/a (see, e.g., Tinte et al, *Phys. Rev. B.* 68, 144105 (2003) and Wang et al, *Nature Materials* 14, 985-990 (2015)).

- 4) The authors also seem to believe that the different magnetic arrangements and linear magneto-electric response they got result from the polarization (via its rotation in the monoclinic phase), as indicated by the sentence on pages 14-15 "We propose that this magnetic arrangement... observed magnetoelectric response". In fact, published papers (Albrecht et al, *Physical Review B* 81, 140401 (R) (2010) and Bellaiche et al, *Journal of Physics Condensed Matter*, 24, 312201 (2012)) demonstrated that these magnetic arrangements and magnetoelectric response are not related to polarization but rather to the tilting of oxygen octahedra. In particular, according to these papers, the weak magnetization in any perovskite material having a G-type predominant magnetic ordering should be aligned along a direction that is both perpendicular to the axis about which the oxygen octahedra tilt and to the G-type antiferromagnetic vector. Such universal law is clearly obeyed in

their (Gy, Fxz) case since G is along the y direction while the axis about which the oxygen octahedra tilt is probably along a [uvw] direction. The authors should correct their mistake but also use such universal law to try to not only explain the results for the (Gxz, Fy) case but also to find the precise direction of the weak magnetization as a function of temperature below 350K, since they should know the (monoclinic and temperature-dependent) direction of the axis about which the oxygen octahedra tilt for any temperature from their structural data in the Cc phase.

- 5) I also found Fig. 4c confusing. How can a linear magneto electric coefficient, alpha, depend on the magnetic field, i.e. how can we have $\alpha = A_0 + A * H$ where A_0 and A are constant? If this is the case, we will have a change in polarization given by $\Delta P = (A_0 + A * H) * H = A_0 * H + A * H^2$. In other words, one would have to 'talk' about a quadratic magnetoelectric coefficient, as well, rather than a linear one only. By the way, the authors may want to replace the comma by period in the vertical axis of Fig 4c giving the values of these magnetoelectric coefficients. The authors may also want to compare their linear magnetoelectric coefficients with those measured or predicted for pure BiFeO3 (see again Albrecht et al, Physical Review B 81, 140401 (R) (2010) and references therein).

Reviewer #3 (Remarks to the Author)

The authors report distinctive room-temperature magnetoelectric responses in a new Bi_{0.68}Pb_{0.32}Fe_{0.64}Mn_{0.04}Ti_{0.32}O₃ ceramic. They experimentally verified the room temperature magnetoelectricity, i.e., the electric field induced phase-change/polarization-rotations and the resulting change in magnetic state.

The most promising point proposed by the authors to enhance the magnetoelectric coupling is the electric field induced phase-change/polarization-rotations effects and the resulting change in magnetic state, giving rising to a large magnetoelectric coupling. However, in order to realize the enhanced coupling using this method, one should have a system with two polymorphic phase coexist that both possess a magnetically ordered phase. Unfortunately, in Bi_{0.68}Pb_{0.32}Fe_{0.64}Mn_{0.04}Ti_{0.32}O₃, the P4mm phase is paramagnetic without net magnetization while the Cc phase is a spin-canted phase at room temperature investigated here. Therefore, it is not an ideal situation to realize the phase-change induced magnetoelectric coupling. It is not reasonable to deduce from this observation that the potential impact of this work has been overstated.

Moreover, there are a number of issues that need to be addressed or further explained.

1) The authors mentioned "a distinctive step-like increase of permittivity at a temperature that shifts with frequency, from 450 K at 1 kHz to 550 K at 1 MHz". However, in Fig. S2, only bumps can be observed, which are too weak to be qualified as a "distinctive" event. This was speculative and overstated. The author should clarify this issue and provide the method to determine the values of those temperatures.

2) The $\text{Bi}_{0.68}\text{Pb}_{0.32}\text{Fe}_{0.64}\text{Mn}_{0.04}\text{Ti}_{0.32}\text{O}_3$ ceramic sample displays a very high remnant polarization ($P_r \approx 78 \mu\text{C}/\text{cm}^2$), one of the largest values ever reported for a ferroelectric ceramics up to date, with a coercive field E_c of 5.4 kV/mm. Such a large value in a likely conductive ceramic looks suspicious of containing electronic/ionic contributions. The authors need to verify the polarization value by calculating the theoretical polarization value based on the displacements of ions obtained from the structural refinements, and explain the origin of high polarization if confirmed.

3) Supplementary Fig. S3 shows direct phase transitions for each polymorph to the high-temperature cubic phase, i.e., the $P4mm$ and Cc show the same phase transition temperature, which is obviously contradictory to the thermal hysteresis observed in Fig. 2. The authors argue that the stress field in the ceramic sample gives rise to the thermal hysteresis. To confirm this, the XRD results of the ceramic are necessary to compare with that of powders, which would provide a clear way to confirm the effects of stress-field on the crystal structure, i.e., the so-called stress-induced phase-change phenomena.

In addition, the detailed phase transition sequences upon heating/cooling in ceramic sample in Fig. 2 need to be specified.

4) The authors mentioned the enhanced spin-canted ferromagnetism. However, the M-H relation is basically linear and no hysteresis can be observed until 50 kOe. The very tiny remnant magnetization could be resulted from existence of the trace amount of strongly magnetic defects, such as Fe^{2+} . Therefore, the statement of the existence of spin-canted ferromagnetism is not convincing, but rather speculative.

5) The authors argued that the Cc phase transforms into $P4mm$ phase after poling, which gives rise to the decreasing divergence between the ZFC and FC curves as well as the decreasing net magnetization. However, to confirm this conclusion, the authors should measure the XRD before and after poling, and compare these two sets of results, which could give a straightforward evidence

for the phase transformation under electric field, i.e. the so-called electric field induced phase-change effects.

Also, it is necessary to discuss whether such an electric field induced phase change is reversible or not.

In conclusion, I cannot recommend this manuscript for publication in Nat. Commun. in its current form.

Reviewer #1 (answers to his/her remarks):

1. Evaluating the effective magnetoelectric response would require the direct characterization of the reversible field-driven phase change for the poled state during subsequent unipolar (parallel to the poling) driving fields. This involves accomplishing diffraction experiments under applied high electric fields (up to ~ 10 kV mm⁻¹ and at 250 K in our case), and the subsequent Rietveld analysis of the patterns to obtain the volume percentage that is reversibly transformed from one polymorph to the other for a given applied electric field. The associated magnetization change could then be calculated.

However, this experiment in bulk materials (contrary to thin films) is currently highly challenging, and requires specifically devoted facilities to perform XRD experiments at the required very high voltages and low temperatures, to which we have no access in the short term. Instead, we have carried out a direct characterization by XRD of the irreversible field-driven phase change upon poling, which is now included in the Supplementary information (see Fig. S5), thus providing direct evidence of the phase-change phenomenon. This was achieved by carrying out a series of XRD experiments on the same ceramic sample before and after poling the material at high electric fields (and 250 K) after the hysteresis loop (described in Page 11).

Note that some stress-driven relaxation after removing the electric field is necessary for a reversible field-induced phase change to take place, which would be the actual source of the effective magnetoelectric response. Note also that additional phase changes during subsequent polarization reversal are not expected, for it is thought to be controlled by 180° switching within the new mixed-phase configuration. Actually, the use of the new material for memory applications would require a procedure to revert the phase change. In mixed-phase BiFeO₃ epitaxial films this was locally achieved by applying a moderate (below the coercive one), opposite electric field (Nature Commun. **2**, 225 (2011)). All these issues have been addressed in the revision.

2. We realize that we have failed to discuss in our original manuscript literature on mixed phase BiFeO₃ epitaxial films (Science **326**, 977 (2009)). Indeed a multiferroic MPB resembling strong analogies to that here reported has been obtained by strain engineering, and a local (using scanning probe techniques) R-like (or Cc) to T-like phase transformation has been demonstrated with a

distinctive effect on magnetism (Nature Commun. **2**, 225 (2011)).

A discussion on the topic has been included in the revision (Page 16).

The advantage of directly obtaining similar effects on a single-phase material without strain engineering, as reported in this work, enables the possibility of using ceramic materials, either bulk or thin films on Pt/Si based substrates. This is clearly advantageous for the potential application of materials (note for instance that commercial FeRAMs are based on polycrystalline PZT films on Pt/Si based substrates). This has been also highlighted in the revision (Page 16).

In relation to the literature on magnetic exchange-mediated magnetoelectricity in BiFeO₃-based heterostructures, making use of the coupled magnetic and ferroelectric/ferroelastic domain configurations within the multiferroic, this was already included in the original version (Ref. 15-18).

3. Discussion has been modified to avoid any confusion between the two different mechanisms giving place to magnetoelectric responses; field-induced phase changes, and transverse lattice softening next to instability of the multiferroic state, and between the former and the spin reorientation phenomenon within the monoclinic phase, at which weak ferromagnetism appears. Specifically, the sentences indicated have been rewritten for clarity (Page 15).

4. The reviewer is right in that the intrinsic (as opposed to phase-change) magnetoelectric effect associated with the monoclinic phase should be larger before poling than after, for its initial percentage is indeed higher. However, samples are ceramics and thus polycrystal, and the spontaneous polarization is randomly distributed in space within the ensemble of grains.

Therefore, a net magnetoelectric response, like the piezoelectric or pyroelectric ones, does not exist in ceramics unless a previous poling step is carried out.

5. The reviewer correctly stresses out the likely similarities between the magnetoelectric response of the "Cc" phase in this work with that of BiFeO₃ films grown on (001)-oriented substrates, which are also monoclinic Cc under compressive strain. We were not aware of the PRL **105**, 037208 (2010) paper, which is extremely relevant for our work. Actually, it anticipates an enhancement of the magnetoelectric coefficient of the R-type phase (monoclinic Cc actually) of BiFeO₃ epitaxial films, on approaching the

strain engineered morphotropic phase boundary. This is basically the mechanism we propose to be responsible of the high magnetoelectric coefficient of the monoclinic phase. Therefore, a text relating it to the current results has been included in the revision (Pages 5 and 16).

We referred to this mechanism in the original manuscript as continuous polarization rotation, influenced by (and making a connection with) the literature on ferroelectric MPBs for high piezoelectric response. However, and also taking into consideration comments by reviewer 2, we realize that this is very confusing in the current context, in which discrete polarization rotation takes place in the field-induced phases changes. Therefore, and to avoid confusions between the two active mechanisms, we have decided to avoid the term continuous polarization rotation and use instead lattice transverse softening (or structural softness as said in the former reference), which is also more rigorous.

This has been corrected all across the revision.

There are no reliable experimental values of magnetoelectric coefficients for BFO films. A comparison of the values here reported for the new material with predicted values for BFO, along with experimental ones for bulk BFO has been included (Page 15).

6. We recognize that we are not in a position to discuss if it would be better to have a R-like or a Cc phase for the phase-change response, which would require a comparison of threshold fields and magnetism that we do not have for the R case. Therefore, the sentence in the text suggesting that Cc might be advantageous has been deleted.

Reviewer #2 (answers to his/her remarks):

1. We thanks to the reviewer for driving our attention to these published works that we unfortunately overlooked in the original manuscript, and that are very relevant. Specifically, the two previous works pointed out have been quoted, and their results discussed in relation to our own results (Page 16). Indeed a paramagnetic to antiferromagnetic transition that is induced by an electric field has been anticipated in Nd substituted BiFeO₃, associated with the field-induced orthorhombic *Pnma* to rhombohedral *R3c* transition.

2. The misleading sentence in the introduction has been rewritten according to the observation made by the reviewer (Page 4). Two new references (10 and 12) have been also included.

3. We have no evidence of chemical decomposition either by Rietveld analysis of high-resolution XRD data on powders, or by EDXS and SEM analysis on ceramics. This has been highlighted in the text (Page 8). Besides, there are a number of previous studies that showed the evolution of the tetragonal distortion along the $\text{BiFeO}_3\text{-PbTiO}_3$ solid solution (J. Mater. Chem. 21, 3125 (2011)). Though it is true that the tetragonal distortion gradually increases on approaching the MPB, giant tetragonality is already found for tetragonal single-phase compositions without any coexisting monoclinic (or rhombohedral component). This does not rule out that a “negative pressure” contribution might exist in the coexistence region, but it is not necessary for the giant tetragonal distortion.

Indeed, the additional expansion of the monoclinic lattice obtained upon poling the ceramic sample, evidenced by a shift to lower 2θ angles of the position of the $(110)_p$ monoclinic peaks (see Supplementary Fig. S5), might be a manifestation of this effect associated with the resulting mixed-phase configuration (Page 13).

4. As already discussed in relation to the 5th comment of reviewer 1, we realize that the usage of the term “continuous polarization rotation” in the current context is extremely confusing. It is not only that it can be mixed up with the discrete polarization rotation involved in the phase-change responses, but it can also suggest a connection between weak magnetism and polarization that does not exist. This has been addressed as already explained (Pages 5 and 16).

Also, a discussion on the relationship between weak ferromagnetism and the antiferrodistortive motions has been included (Page 15), and its implications in the spin reorientation transition addressed. We have calculated the crystal polarization from the ion displacements provided by the structure refinements for both monoclinic and tetragonal symmetries, along with the direction of the polarization in both cases. Results have been included in the Supplementary information (see Fig. S5).

About the possibility of calculating the octahedra rotation angle, it is true that in a octahedra rigid-model

of the structure, the axis about which the oxygen octahedra tilt is along a [uvw] direction due to group theory relationships, i.e., with two equal rotations along the $\langle 100 \rangle$ and $\langle 010 \rangle$ pseudo-cubic axes, and another component with a different angle along the $\langle 001 \rangle$ pseudo-cubic axis, as indicated. However, distortions of oxygen octahedra are also allowed by symmetry, and in this compound they are so high and prevalent that oxygen atom positions are not resulting from a simple rotation to which an actual rotation axis can be associated.

5. Confusion most probably arises from our failure to explain the experiment in the text main body (or in the figure caption). The magnetoelectric coefficient is obtained as the ratio between the voltage developed (response) and an applied low amplitude alternate (to enable lock-in detection) magnetic field. The amplitude and frequency of the stimulus are 10 Oe and 10 kHz, respectively. The magnetic field in the x-axis of Fig. 4c is not this low field alternate magnetic stimulus, but a static increasingly high magnetization field. The curve is then how the magnetoelectric coefficient increases, as the total magnetization increases (the set-up does not allow to reach saturation). Linearity was checked by measuring the low-field response as a function of the amplitude of the stimulus (up to 10 Oe) at constant magnetization field. Figure and text referring to it have been modified accordingly (Page 18). Besides, a comparison of the experimental coefficients with predicted and experimental ones for BiFeO_3 has been included as requested (Page 15).

The comma has been replaced by dots in the vertical axis of Fig. 4c as indicated.

Reviewer #3 (answers to his/her remarks):

We agree with the reviewer that ideally one would prefer to have field-driven changes between two ferromagnetic phases, instead of between a weak ferromagnetic and a paramagnetic one. However, we do not think we have overstated the potential impact of our work. Our case is analogous to that described in BiFeO₃ epitaxial films, in which a MPB between monoclinic *Cc* and tetragonal *P4mm* phases is strain engineered (Science **326**, 977 (2009), Nature Commun. **2**, 225 (2011)). However, we have obtained analogous effects in a single-phase material, without the need of epitaxial thin-film technology and strain engineering. This enables the possibility of using polycrystalline materials, either bulk or thin films on Pt/Si substrates, which is clearly advantageous for their potential applications.

1. We are not sure what the reviewer means in relation to Fig. S2a, and what bumps he/she is referring to. The Maxwell-Wagner type dielectric relaxation is very distinctive in the data from our point of view (the step-like increase of permittivity at a temperature that shifts with frequency in the range between 475 and 650 K). One possibility might be that the logarithmic scale used for the permittivity, which somehow smoothen the steps, was misleading. We have included the same plot but in a linear scale (*inset*) to highlight the steps related to the M-W process for clarity. The positions of the steps at increasing frequency are obtained from the inflection points (defined as the maxima in the derivatives).
2. We are aware of the growing concern about the actual ferroelectric origin of many loops reported for multiferroics, which we guess it is behind the reviewer's concern. However, loops here presented are compensated from linear polarization and conduction contributions, by subtracting the response of a parallel RC component that correctly described the low field response. This is a standard procedure, and we are confident that the compensated loops are free of artifacts. We acknowledge the necessity of illustrating this, so the compensation process is now provided in the *Supplementary Fig. S3*.

The spontaneous polarization has been calculated from the ion displacements provided by the crystal structure refinements, as requested. Results are also included and discussed in the supplementary information (see *Supplementary Note S2* and *Table S3*). Values are fully consistent with the phase-change mechanism proposed. This has been also discussed in the revision (Page 14).

3. The XRD diffraction pattern for a ceramic is compared with that of the powder in a new figure included in the supplementary information (Fig. S5). The different percentages of tetragonal and monoclinic phases are evident, which is an effect of ceramic stress on the phase coexistence. The temperature evolution of the phases in powdered samples clearly shows differentiated transition temperatures for monoclinic and tetragonal polymorphs (Supplementary Fig. S5b), so as the transition of the monoclinic polymorph to the high-temperature cubic one takes place before that of the tetragonal one, in a good agreement with the permittivity data indicating a large thermal hysteresis. We do not think that XRD experiments in ceramics, whose interpretation will be more ambiguous than that for powders, will add up nothing to the main conclusions of the article.

4. Fe_{Fe} point defects are indeed present in the perovskite, and they most probably play a role in the magnetic properties, besides in the electrical ones (hopping conductivity and ferroelectric hardening). However, we have no evidences of traces of secondary phases, so the point defects and any role in the magnetic properties are characteristic of the perovskite oxide here reported. In the case of the spin reorientation phenomenon, the ability of tuning it along the solid solution demonstrates it (J. Mater. Chem. C **3**, 2255 (2015) and Supplementary Fig. S4).

The M-H relation is basically linear due to the strong AFM behavior of the system, and the canted-AFM character of these materials is well recognized. The hysteresis and remnant magnetization can be even obtained at low magnetic fields. Indeed, the enhanced spin-canted ferromagnetism below the spin reorientation is apparent firstly from the comparison of the ZFC/FC curves of both samples (the ceramic under study and the binary system $\text{BiFeO}_3\text{-PbTiO}_3$).

5. A series of XRD diffraction experiments were performed on the ceramic sample before and after poling the material at high electric fields (and 250 K) as requested. Results have been included in the supplementary information (see Supplementary Fig. S5), which provides a direct evidence of the field-driven phase change phenomenon. Reversibility requires carrying out XRD measurements under high electric fields, which are not straightforward as explained in relation to the first comment of reviewer 1.

Reviewers' Comments:

Reviewer #1 (Remarks to the Author)

The authors have made a serious effort to respond to my review, and my criticisms have been convincingly addressed for the most part. The interest of having in a ceramic the same kind of multiferroic effects found in films is now more clear, both in the text and in my mind. Further, the provided evidence on the phase-change transformation is convincing. It would certainly be better to have a direct characterization, but I understand this is a difficult experiment and it makes sense to leave it for future work (which I strongly encourage the authors to pursue).

Unfortunately, the authors did not copy the comments of the other referees in their response, and I have not been able to find the other referees' reports in the web server; hence, I cannot judge whether the authors' answers to the remarks from the other referees are satisfactory.

At any rate, as far as I am concerned, I recommend this paper be accepted for publication in Nature Communications.

Reviewer #2 (Remarks to the Author)

The authors have very well addressed the detailed and numerous comments of the three Reviewers, in my mind. Moreover, the discovery of a system possessing a coexistence of two different ferroelectric phases (including one of low-symmetry and having a small magnetization), and possessing (linear) magneto electricity at room temperature is an important finding, both fundamentally and technologically.

As such, I would like to recommend this manuscript for publication in Nature Communications.